# RhoA Signaling in Immune Cell Response and Cardiac Disease

**DOI:** 10.3390/cells10071681

**Published:** 2021-07-03

**Authors:** Lucia Sophie Kilian, Derk Frank, Ashraf Yusuf Rangrez

**Affiliations:** 1Department of Internal Medicine III, Cardiology, Angiology, Intensive Care, University Medical Center Kiel, 24105 Kiel, Germany; lucia.kilian@uksh.de; 2DZHK, German Centre for Cardiovascular Research, Partner Site Hamburg/Kiel/Lübeck, 24105 Kiel, Germany; 3Department of Cardiology, Angiology and Pneumology, University Hospital Heidelberg, 69120 Heidelberg, Germany

**Keywords:** RhoA, cardiac inflammation, immune cells, cardiocrine signaling, cardiac diseases

## Abstract

Chronic inflammation, the activation of immune cells and their cross-talk with cardiomyocytes in the pathogenesis and progression of heart diseases has long been overlooked. However, with the latest research developments, it is increasingly accepted that a vicious cycle exists where cardiomyocytes release cardiocrine signaling molecules that spiral down to immune cell activation and chronic state of low-level inflammation. For example, cardiocrine molecules released from injured or stressed cardiomyocytes can stimulate macrophages, dendritic cells, neutrophils and even T-cells, which then subsequently increase cardiac inflammation by co-stimulation and positive feedback loops. One of the key proteins involved in stress-mediated cardiomyocyte signal transduction is a small GTPase RhoA. Importantly, the regulation of RhoA activation is critical for effective immune cell response and is being considered as one of the potential therapeutic targets in many immune-cell-mediated inflammatory diseases. In this review we provide an update on the role of RhoA at the juncture of immune cell activation, inflammation and cardiac disease.

## 1. Introduction

Since Levine et al. reported elevated serum levels of tumor necrosis factor (TNF) in heart failure (HF) patients, evidence is accumulating that activation of proinflammatory pathways and chronic inflammation play an important role in HF [1,2]. For example, inflated levels of several pro-inflammatory cytokines like TNF-α, interleukin (IL)-1 and IL-6 have been observed in and associated with HF prognosis [2,3,4]. Mechanistically, increased expression of TNF-α and IL-1β leads to cardiomyocyte hypertrophy, likely through induction of transcription factors such as nuclear factor-κB (NF-κB) [5,6,7]. Generation of a pro-inflammatory milieu characterized by secretion of cytokines and chemokines further stimulate migration of monocytes and their differentiation into macrophages in the cardiac tissue [8,9,10]. In turn, the expansion of immune cells promotes pathological tissue remodeling and induces cardiomyocyte apoptosis and ultimately cardiac decompensation [11].

The immune system is based on two main complexes: (1) the innate immune system and (2) the adaptive immune system. The innate immune system is triggered by direct contact with pathogens or inflammatory/danger signals and includes non-cellular responses, i.e., the release of inflammatory cytokines, and cellular responses, i.e., infiltration of innate immune cells (macrophages, dendritic cells, and granulocytes) into affected tissue [12]. The adaptive immune system also involves non-cellular (hormonal) and cellular responses (stimulation of B- and T-cells), but in contrast to the innate response, it builds up an “immunologic memory” by developing pathogen-specific receptors [12]. RhoA (ras homolog family member A), an ubiquitously expressed small GTPase, acts as a molecular switch not only in the activation of cytoskeletal proteins but also in responding chemokines, cytokines, and growth factors released from both innate and adaptive immune cells [13]. Furthermore, RhoA activation and RhoA-dependent signaling pathways in cardiomyocytes [14] and in immune cells [13,15] have been shown to mediate immune responses, which play an important role in pathogenesis and progression of cardiac dysfunction [12]. Present review aims at summarizing the possible role RhoA plays in cardiac disease via mediating the host immune system and how it can be exploited therapeutically to improve cardiac health of patients.

## 2. Links between Cardiac Hypertrophy, Heart Failure, and Immune Cell Activation

In the last decades, the role of the innate and adaptive immune response is being linked with a number of signaling molecules and pathways, including RhoA activation in cardiomyocytes [13,16]. Furthermore, it has been illustrated that different cardiac diseases (e.g., ischemic, hypertensive, and genetic cardiomyopathies) converge in inducing a common immune response that contributes to disease progression [12]. However, exact causes, underlying mechanisms and signaling pathways of the interplay between cardiac diseases, and pro- as well as anti-inflammatory immune system responses, are still largely unknown.

Signals that trigger activation of innate immune cells and subsequent immune responses are pathogen-associated molecules or “danger-molecules” and other signals arising from damaged tissue [12]. The main group of receptors for inflammatory signals consists of pattern recognition receptors (PRRs) [12]. PRRs, which are generally upregulated in HF [17], are commonly expressed by immune cells, but have also been found in cardiac cells [18,19]. Although PPRs are best known for their activation after contact with pathogens, it can also be activated by danger-associated molecular patterns (DAMPs) that are released by damaged or dying cells [20], e.g., injured myocardium. The activation of PPRs triggers the release of inflammatory cytokines [21] (e.g., interferones (IFN) and IL-1β) and thus accelerates immune responses [12] (Figure 1).

Natriuretic peptides are key players of cardiac hypertrophy and HF [14]. Brain and atrial natriuretic peptides (BNP and ANP) are part of the “hypertrophic gene program” in cardiomyocytes and their upregulation, which is partly mediated by RhoA activation, is an established marker for cardiac hypertrophy [14,16,22]. In recent years, the function of natriuretic peptides as cytokines and the links between cardiomyocytes and immune cells have come into focus [23]. Immune cells, especially macrophages, have been shown to produce natriuretic peptides and express their receptors [23]. In vitro experiments demonstrated that inflammatory cytokines like TNF-α and IL-1β upregulate BNP in cardiomyocytes [24], whereas ANP can reduce the activation of NF-κB, thus reducing TNF-induced infiltration by neutrophils [25,26,27]. So, the upregulation of “hypertrophic gene program” might induce a feedback loop attenuating inflammation [12]. However, more studies are needed to understand the links between hypertrophic signaling pathways in cardiomyocytes and immune cell responses.

In addition, different forms of programmed cell death have been associated with the development and progression of cardiac diseases [21,22]. Besides apoptosis, this includes “programmed necrosis” or necroptosis of myocardial cells [21,22,23]. Cardiomyocyte necroptosis has recently been evidenced in patients with dilated cardiomyopathy (DCM) and ischemic heart disease [28,29]. Along these lines, experiments with isolated rodent hearts have shown cardioprotective effects of necroptosis inhibitors [30,31]. On the other hand, TNF-α promotes necroptosis via binding to TNF-receptor 1 (TNFR1), and initiates a signaling cascade including the activation of receptor interacting protein kinase 1 (RIPK1), receptor interacting protein kinase 3 (RIPK3), and mixed lineage kinase domain-like protein (MLKL) [32,33]. Downregulation of TNFR1 in a mouse model of cardiac hypertrophy reduced myocyte hypertrophy, inflammation, and fibrosis induced by transverse aortic constriction (TAC) [34]. Interestingly, Sharifi et al. recently described a link between cardiomyocyte necroptosis and RhoA signaling [35]. They showed that in a mouse model of myocardial ischemia-reperfusion (MI/R) that the induced fibrosis, cardiac tissue damage, and heart dysfunction is associated with increased RIPK1-RIPK3-MLKL and RhoA signaling [35]. Whereas, treatment with nesfatin-1 reduced the expression of RIPK1, RIPK3, MLKL and the RhoA effectors of Rho-associated, coiled-coil-containing protein kinases 1 and 2 (ROCK1 and ROCK2), as well as attenuated the morphological and functional cardiac damage [35].

The best studied immune cells in the context of cardiomyopathies are macrophages. In general, two classes of macrophages are defined: (1) classically activated macrophages (=M1), which correlated with pro-inflammatory processes and (2) alternatively activated macrophages (=M2) correlating with anti-inflammatory processes. Most macrophages found in damaged (cardiac) tissue originate from myeloid naïve monocytes, which circulate in the blood stream, migrate into affected tissue, and differentiate after activation, which belong to the M1 class [36]. In addition, one subclass of macrophages, called resident macrophages (rcMACs), resides in specific tissues including myocardium under physiological circumstances [36,37]. These residential cardiac macrophages can get activated directly by signals of surrounding (cardiac) cells and they belong to the M2 class [36,37]. In this context RhoA signaling has been linked to macrophage polarization via its main effector, ROCK1, in mice cells [38]. Furthermore, activation of macrophages has been demonstrated to correlate with the progression of cardiac hypertrophy and fibrosis to HF [12,39,40,41] (Figure 1). For heart failure with preserved ejection fraction (HFpEF), which is initially characterized by hypertrophy followed by cardiac fibrosis and left ventricular dysfunction, the involvement of (innate) immune responses, and especially, activation of macrophages have been described [12]. Cardiac (resident) macrophages have been shown to induce cardiomyocyte death and fibrosis during HFpEF [42]. In addition, enhanced numbers of monocytes in the blood and activation of M2-type macrophages have also been shown to correlate with left ventricular dysfunction and progression of HFpEF [43]. DCM caused by congenital mutations and manifested as dysfunction of the left ventricle has also been associated with activation of immune responses [39].

It has been suggested that pro-inflammatory signals in hypertrophic hearts generated by injured myocardium induce differentiation of circulating monocytes to pro-inflammatory M1-type macrophages, and even induce pro-inflammatory responses in residential macrophages, enhancing disease progression [39,40]. In addition, an increase in M2-type macrophage polarization in the heart has been linked to progression of hypertrophy to HF, because the anti-inflammatory responses are known to induce fibrosis [12]. However, treatment with the anti-inflammatory cytokine IL-10 in mice has been shown to block cardiomyocyte hypertrophy and fibrosis induced by pressure overload or hormonal stimulation [44]. Overall, inflammation in cardiac tissue correlates with higher numbers of macrophages in the cardiac tissue, but the origin of these macrophages (resident or infiltrating), their polarization (M1 or M2) and their effect (pro or contra disease progression), as well as the exact role of RhoA signaling is still under discussion.

Besides macrophages, dendritic cells (DCs) and neutrophils are the main actors of the innate immune response system. DCs are derived from myeloid progenitors and are located in tissues with contact to the external environment and in the blood. Upon activation, they migrate to lymph nodes, generate and present specific antigen complexes and so activate B- and T-cells. DCs have been found in cardiac tissue injured by cardiac infarction and there is some evidence that they are involved in the subsequent cardiac remodeling [45,46]. In this context, a beneficial effect of DC activation in the heart by regulation of monocyte/macrophage homeostasis has been suggested [46]. In contrast, DCs might also promote the development and progression of cardiac hypertrophy and HF [47,48]. Recruitment of T-cells, which is generally induced by stimulation of DCs, has been shown to promote the transition from cardiac hypertrophy to HF in a mouse model of pressure-overload by TAC in mice [47,48]. However, no detailed studies on the activation and function of DCs in cardiac hypertrophy are published up until now. Neutrophils are also myeloid cells and the most abundant class of granulocytes. They are best known for their role in anti-bacterial responses. Moreover, neutrophils which are located in blood vessel walls can become activated by biomechanical (shear-)stress, and in response to inflammatory signals, migrate into the affected tissue, and release more inflammatory signals inducing cell death and/or co-stimulation of other immune cells [12,49]. There is some evidence that neutrophils also play a role in cardiac remodeling after infarction, as depletion of neutrophils in mice reduced reperfusion-induced necrosis [50]. However, elaborate research on the role of neutrophil activation in cardiomyopathies is still missing.

Data on links between cells of the adaptive immune system (T- and B-cells) and cardiac diseases, especially hypertrophy, is also insufficient. As mentioned above, there is some evidence that recruitment of T-cells enhances the progress of TAC-induced cardiac hypertrophy [47,48]. Furthermore, mice with depletion of T-cells exhibit lower number of infiltrating macrophages, reduced fibrosis, and reduced cardiac dysfunction after TAC [47]. Nevertheless, more studies are needed to identify interconnections of hypertrophic cardiomyocytes and activation of T- and/or B-cells. Taken together, there is some evidence for links between cardiac hypertrophy and immune system responses. An overlap of risk factors for cardiomyopathies and (chronic) inflammation as well as the activation of immune cells in a variety of cardiac diseases are promising fields for further research, as is the postulated interplay of upregulation of hypertrophic markers (BNP and ANP) and cytokine signaling [23,24,25,26,27].

## 3. RhoA Activation and Signaling in Immune Cells

Recent studies demonstrate that RhoA plays an important role in immune responses, with its effects being highly dependent on the spatio-temporal regulation of RhoA activation in different innate and adaptive immune cells [13,51]. On molecular level, the activation of RhoA in immune cells and cardiomyocytes depends on its change from a guanosine diphosphate (GDP)-bound “inactive” to a guanosine triphosphate (GTP)-bound “active” state and back [52]. This cycle is regulated by regulatory proteins, namely guanine nucleotide exchange factors (GEFs), GTPase activating proteins (GAPs), guanine nucleotide dissociation inhibitors (GDIs) and GDI dissociation factors (GDFs) [16]. GEFs facilitate the dissociation of GDP from RhoA, thus accelerating the binding of GTP and allowing the activation of downstream effectors [53]. GAPs catalyze the hydrolysis of GTP back to GDP and thus the release of the effector [44]. GDIs disconnect RhoA from the plasma membrane, thus inhibiting the dissociation of GDP [54], while GDFs initiate the dissociation of GDIs from RhoA allowing the cycle to start again [16]. A number of these regulators have been shown to be identical in immune cells and cardiomyocytes (e.g., LARG (leukemia-associated RhoGEF), GEF-H1/Lfc (Lbc’s first cousin), PDZ-RhoGEF, p190RhoGAP, and Vav1) [49,55], while others were found specifically in immune cells (e.g., RhoA-GAP, Myo9B, and Rho-GEF7) [56]. Even some of the up- and downstream signaling pathways of RhoA described in cardiomyocytes have been proven to play an important role in immune cells, e.g., RhoA activation via Gα12/13-coupled membrane receptors [57,58,59] and RhoGEF or activation of the RhoA effector, ROCK [13,15,57,58,59,60].

RhoA activation has also been associated with activation of β2-adrenergic receptors (β_2_-AR), probably via p115RhoGEF [61,62]. Hyperactivity of the sympathetic nervous system is one of the hallmarks of heart failure that involves catecholamine spillover, and is associated with pro-inflammatory signaling [63,64,65]. Several studies have shown that infusion of isoprenaline—a synthetic catecholamine and β_2_-AR agonist—in mice induces cardiac inflammation and dysfunction [66], and infusion of noradrenalin, also a β_2_-AR agonist, induces cardiac hypertrophy and fibrosis in rats [67]. However, β_1_-ARs have been shown to be ubiquitously expressed in rodent (ventricular) cardiomyocytes, while β_2_-ARs were only found in a very small percentage of these myocytes [68]. In contrast, β_2_-ARs were found to be abundant in non-myocytes of rodent heart tissue [68], suggesting that endothelial cells, fibroblasts and/or immune cells in cardiac tissue likely express β_2_-ARs.

Overall, the processes affected by RhoA in the context of immune responses are fundamental cellular processes such as cytoskeletal arrangement, mobility, cell-cell contact, cell-cycle regulation, proliferation and cell survival, as well as directed migration and (co-) stimulation of immune cells (Figure 2 and Figure 3).

### 3.1. RhoA Activation in Macrophages, Dendritic Cells, and Granulocytes

In macrophages, DCs, and granulocytes, it has been shown that, in general, RhoA activation in the rear end (uropod) [15] and RhoA inhibition in the leading edge (podosome, pseudopod) [15,56] are essential for directed migration (Figure 2A–C). However, a number of experiments have demonstrated that the spatio-temporal regulation of RhoA activation in the immune cells is essential for effective immune responses [13,51].

At the leading edge of macrophages, RhoA is inhibited by the Rho-GAPs Myo9B (myosin XIB) and p120RhoGAP, which in turn are up-regulated by Krüppel-like factor-5 (KLF-5) and the B-cell lymphoma-6 (Bcl-6) molecule via the macrophage colony stimulating receptor 1 (CSF1), respectively [69]. Consistently, knockout of Myo9B as well as inhibition of Bcl-6 led to upregulation of active RhoA in macrophages [70,71]. In accordance with the importance of spatio-temporal regulation of RhoA, the directed migration of the macrophages was impaired in both experiments [70,71]. The activation of RhoA at the uropod was stimulated by pro-inflammatory cytokine tumor necrosis factor alpha (TNF-α)-induced signaling via PI3K (phosphoinositid-3-kinase) and the Rho-GDF ERM (ezrin/radixin/moesin) [72]. RhoA activated the downstream effectors ROCK and mDia facilitating the retraction of the rear end [73]. Treatment with the anti-inflammatory cytokine TGF-β1 (transforming growth factor beta) first induced the RhoA activation at the rear end, but over time, inhibited it via activation of protein kinase A (PKA) and p190RhoGAP [74] (Figure 2A).

RhoA activation in the rear end of DCs with the consecutive activation of ROCK and diaphanous-related formin-1 (mDia), as well as RhoA inhibition by Myo9B have also been suggested as essential signaling pathways for their cytokine-induced directed migration [75,76]. Myo9B knockout mice had lower levels of active RhoA in the DCs and impaired directed migration, similar to the findings in macrophages [70,77]. RhoA and mDia have been found to be enriched in the rear end of immature motile mice DCs in vitro [75], and in vivo inhibition of ROCK in mice skin DCs impaired their migration [78]. Likely time-dependent activation of RhoA and crosstalk with other Rho-family protein signaling pathways also play a role here, as another small GTPase of the Rho/Rac/Cdc42 family, Cdc42 (cell division control protein 42 homolog), was enriched in the front end of mice DCs in vitro [75]. Knockout of the RhoA-inhibiting and SWAP-70 (switch-associated protein 70), an activator of the third member of this family of small GTPases, Rac [79], interfered with the distribution of the small GTPases of Cdc42, Rac, and RhoA and led to an increased RhoA level, which impaired directed migration and attenuated retraction of the rear end in response to cytokine stimulation [80]. However, DCs with knockout of the Rho-GEF5 and consequently reduced RhoA level also exhibited impaired migration in vitro and in vivo [76] (Figure 2B).

In neutrophils, active Cdc42 and Rac were found at the leading edge and activated RhoA at the rear end, too [51,81]. Worthylake et al. have demonstrated that RhoA signaling is essential for retraction of the rear end and thus monocyte transendothelial migration, as monocytes treated with the RhoA-inhibitor C3-transferase showed initial but incomplete forward and transmigration movement, with deficient retraction [81]. Isolated human neutrophiles treated with a ROCK inhibitor have been shown to have impaired force generation and disruption of the asymmetric force pattern leading to reduced motility [82]. Upregulation of RhoA in the rear end of neutrophils was associated with upregulation of RhoGEFs (e.g., PDZ-RhoGEF, LARG, p115RhoGEF/Lsc (Lbc’s second cousin), and GEF-H1/Lfc) [55,57,83]. In human leukemia cells (HL-60 cell line), impairment of PDZRhoGEF led to abnormal cell polarization [57]. In murine neutrophiles, the RhoA-GEF LARG is found to be upregulated during directed migration via mDia-dependent signaling [55,84]. Furthermore, neutrophiles of mice lacking the RhoA-GEF Lsc show undirected migration due to deficient pseudopod formation [57]. The RhoGEF activation was shown to be, in part, linked to Gα12/13-coupled receptor activation and led to RhoA-induced accumulation of ROCK and mDia, which is needed for retraction of the rear end and directed migration [55]. Besides, there is some evidence for compensatory functions of the RhoA-GEFs, as neutrophils of the Lsc-deficient mice showed normal directed migration to inflammatory sites in vivo [57]. RhoA signaling has emerged as an important factor in rear-front signaling for asymmetric force generation in neutrophils as well as an essential factor for maintenance of neutrophile polarity and efficient migration [51]. In this context, a mechanism including different small GTPases, namely Rho, Rac, and Cdc42 at the rear and front edge as well as feedback loops has been described [51]. Actin polymerization and Rac and Cdc42 activity is prominent at the leading edge and affects downregulation of RhoA, while contractile actin-myosin filaments are associated with high RhoA activity at the trailing edge and affect downregulation of Cdc42 and Rac [85,86,87]. Feedback loops reinforce the spatially different expression of the small GTPases, generating gradients of RhoA, Rac, and Cdc42 activity from between the rear and front edge, and thus sustain the polarity of neutrophils [87]. However, this complex signaling network is not completely understood yet, and while all three GTPases definitely play some part in the extent and mechanisms by which these small GTPases regulate neutrophil motility, the signaling network is still under discussion [13,51,87].

### 3.2. RhoA Activation in T-Cells and B-Cells

The effect of RhoA expression in cells of the adaptive immune system is also highly controlled in a spatio-temporal manner [13,88]. Pollock et al. have shown that RhoA-induced ROCK/mDia activation in the uropod of T-cells is essential for their directed migration, as in innate immune cells [89] (Figure 3). Similarly, inhibition of RhoA by the Rho-GAP Myo9B in the leading edge was also shown to be needed for directed migration of T-cells [90]. Consistently, depletion of Myo9B in T-cells lead to increased RhoA activation causing impaired directed migration, as described for innate immune cells, and attenuated protrusion through extracellular matrix [90]. However, other in vitro experiments proved that time-dependent regulation of RhoA activation in T-cells is essential.

In addition to migration, RhoA has been shown to play an important role in T-cell-associated (co-)stimulation of immune cells [91]. The knockout of the RhoA-GAP Myo9B in mice, enhancing RhoA activation, did not only impair migration of innate and adaptive immune cells, but also affected DC/T-cell contact and reduced T-cell proliferation [90]. Knockout of SWAP-70 in DCs, also enhancing RhoA activation, led to reduced expression of the pro-inflammatory CD4^+^ (cluster of differentiation 4 positive)-T-cell stimulatory surface marker MHC-II (major histocompatibility complex class II molecules) [92], but increased presentation of the anti-inflammatory CD8^+^-T-cell stimulatory marker MHC-I (major histocompatibility complex class I molecules) [93]. Consistently, mice overexpressing p190RhoGEF, which is associated with increased RhoA activation, showed defective co-stimulation capacity of T-cells [80] (Figure 3). RhoA activation is associated with T-cell receptors (TCRs), which recognize MHC/antigen complexes presented at the surface of other cells and induce a variety of downstream signaling pathways, leading to co-stimulation of other immune cells and enhancement of pro-inflammatory signaling [13,88]. Mice with deficient RhoA-expression in T-cells show reduced proliferation and even affected their survival [88,94]. However, Rho-GEF Vav1 was shown to get activated by TCR activation and function as part of a negative feedback loop, reducing the TCR-induced co-stimulation [95,96]. In addition, RhoA was shown to compromise the expression of pro-inflammatory and co-stimulating cytokine IL-2 by T-cells [97]. 

Like its function in T-cells, RhoA activation in B-cells has been linked to B-cell receptor (BCR) activation and B-cell proliferation [98]. RhoA downregulation attenuated B-cell proliferation induced by BCR-activation [98]. Furthermore, B-cell activation depends on CD4^+^-T-cell activation and MHC-II presentation (Figure 3). On the one hand this activation has been associated with increased RhoA expression via p190RhoGEF [99]. On the other hand, as described above, RhoA has been suggested to induce downregulation of CD4^+^/MHC-II stimulation [92]. This might represent another negative feedback loop.

Taken together, in vitro and in vivo experiments have demonstrated that RhoA is a key player of immune cell responses, influencing the migratory and stimulatory capacity of innate and adaptive immune cells [13,51,88]. In general, RhoA upregulation in the rear end and RhoA inhibition in the leading edge of innate and adaptive immune cells is necessary for their directed migration. However, an important aspect is the spatially and temporally strictly controlled regulation of RhoA activation to allow effective immune responses. Emerging evidences also suggest that the Rho-GTPase signaling pathways crosstalk with each other and are influenced by cellular mechanics, leading to self-organization of the several dynamic cellular processes [100,101,102]. Further research is thus needed to unravel the complex regulation and effects of RhoA activation in immune cells and the impact on immune responses linked to cardiac diseases.

## 4. Perspectives

The above-described correlations between immune cell activation and the development and progression of cardiac diseases are promising starting points for further research.

By analyzing the up- and downstream pathways of RhoA activation in different cells of the innate and adaptive immune system, more mediators of RhoA signaling could surely be identified and more regulatory mechanisms of immune cell responses can be described. Important in this context is to pay heed to the spatio-temporal regulation in order to gain valid and comparable data (e.g., before or after final differentiation or infiltration and localization at the rear or front side).

Another interesting approach to gain more insight into the role of RhoA in the development and progression of cardiac diseases and immune system responses would be large-scale screens for correlations between mutations in the human RhoA-gene and the development of cardiac and/or immune cell-related diseases. Up until now, there are no RhoA mutations corresponding with cardiac diseases known yet, but there is good evidence for direct correlations between mutations of the RhoA-gene as well as known mediators of RhoA signaling pathways and cancer. The mutation RhoA-G17V (a missense mutation leading to the change of the amino acid glycine on position 17 to valine) is common in CD4^+^ T-cells in lymphoma [103] and is associated with autoimmunity, hyper-responsiveness of T-cells, and excessive TCR signaling via the Rho-GEF Vav1 [104]. Moreover, a gain-of-function mutation of Vav1 itself has been shown to be over-represented in T-cell lymphoma [104].

Vav1 inhibition using dasatinib, which is already used against leukemia [105], reduces TCR-stimulation in lymphoma cells [104]. Accordingly, Vav1 inhibition using dasatinib is a promising tool to impair RhoA activation in cardiomyocytes and immune cells, thus attenuating inflammatory processes which promote the progression of cardiac dysfunction. In addition, statins are known to be effective inhibitors of RhoA-dependent signaling [106], and further research could lead to new therapeutic approaches for cardiac diseases in this context.

Another promising approach for cardioprotective regulation of RhoA/ROCK signaling is the reduction of necroptosis, by interfering with, e.g., through TNFR1 and the RIPK1-RIPK3-MLKL-axis [30,31,34,35]. This certainly deserves further research, taking into account that TNFRs can be expressed by cardiomyocytes as well as immune cells [107,108,109].

Furthermore, beta-blockers, which function as antagonists for ARs, have long been established for the treatment of patients suffering from heart failure [110,111]. However, specificity is important in this context. While the main effect of beta-blockers lies in their inhibition of β_1_-AR activation, thus regulating heart rate and improving contraction, many beta-blockers are non-selective and also inhibit probably cardioprotective signaling via β_2_-AR [112]. Therefore, more studies analyzing the links between β-ARs and RhoA activation are needed, while subtype selectivity of β-AR regulators (e.g., β_1_-AR inhibitors and β_2_-AR stimulators), as well as probable cell-type specific effects have to be taken into account to improve therapeutic approaches.

Overall, specificity is the key factor in all approaches using inhibitors, and more research is needed to analyze potentially beneficial effects of RhoA inhibition in immune cells and cardiomyocytes, and to detect off-target effects, for example, on Rac or Cdc42.

In addition, further analysis of the association of the release of danger signals by injured, or even just stressed, cardiomyocytes with subsequent cytokine-induced immune cell activation might lead to the identification of new circulating molecules, which can be used as markers for the extending of myocardial stress and damage.

## 5. Conclusions

Taken together, the existing data on the interplay of cardiomyocytes and immune cells demonstrates good evidence for links between cardiomyocyte damage, immune cell activation and progression of cardiac dysfunction. The intercellular signaling involved in and affected by the development and progression of cardiac diseases includes cardiomyocytes, cells of the innate and adaptive immune system, and other cells present in the myocardium (e.g., fibroblasts). The summary of the data has pointed out that in the intracellular signaling, RhoA is one of the key mediators. Regulation of RhoA activation and RhoA-dependent downstream signaling plays an important role in cardiomyocytes and immune cells. The data on effects of RhoA activation and inhibition is often contradictory, depending on specific cell types, spatial, and temporal regulation. For further research, induction of immune cell responses should be considered when analyzing the mechanisms of the development and progression of cardiomyocyte damage and cardiac dysfunction.

## Figures and Tables

**Figure 1 cells-10-01681-f001:**
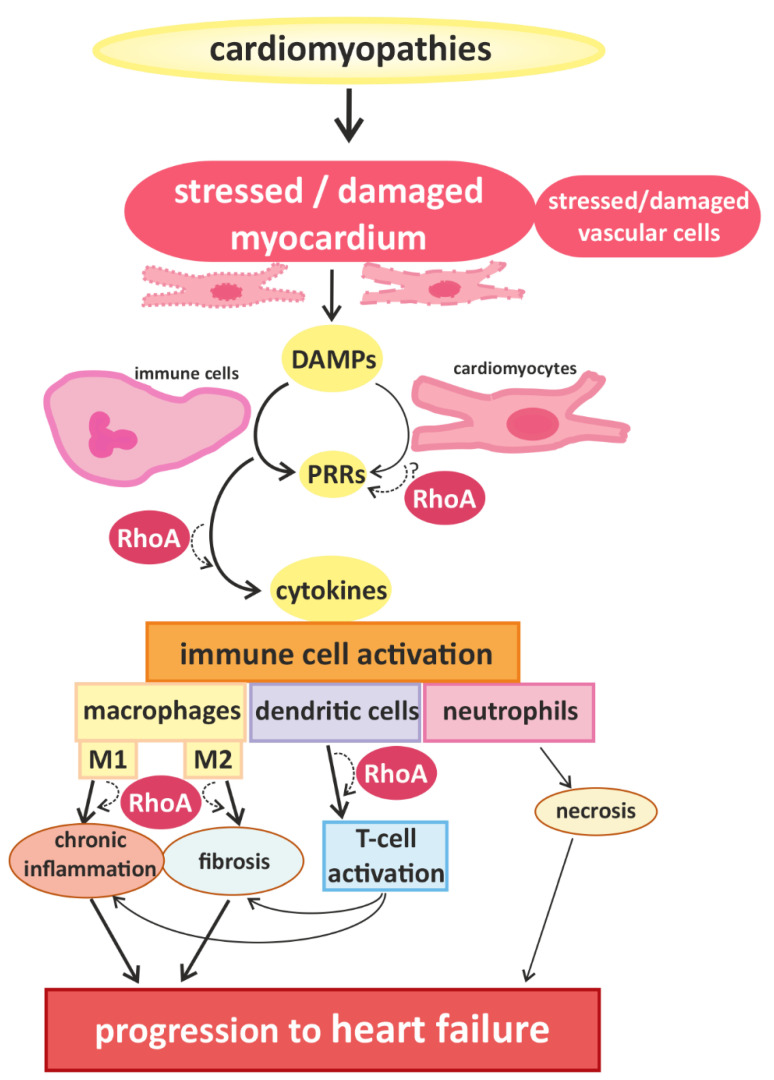
Cardiomyopathies of different causality converge in myocardial damage, inducing immune cell responses and promoting disease progression. Cardiomyocytes and blood vessel cells that are stressed or injured release danger signals with “danger-associated molecular patterns” (DAMPs), which are recognized by “pattern recognition receptors” (PRRs), expressed by immune cells and cardiomyocytes. The activation of PRRs induces the migration and activation of (more) immune cells via cytokine signaling. In damaged myocardium, classically activated M1-type and alternatively activated M2-type macrophages are found, which are associated with chronic inflammation and fibrosis, respectively. Furthermore, dendritic cells and co-stimulated T-cells contribute to the development of inflammatory and fibrotic remodeling. Infiltrating neutrophils are associated with increased necrosis. These processes accelerate the progression of cardiac dysfunction. RhoA-dependent signaling is essential for effective immune cell activation and might also play a role in the transduction of danger signals and receptor activation in cardiomyocytes. In addition RhoA signaling is involved in macrophage polarization and signaling pathways mediating the interaction between dendritic cells and T-cells, contributing to pro- and anti-inflammatory remodeling and finally, heart failure.

**Figure 2 cells-10-01681-f002:**
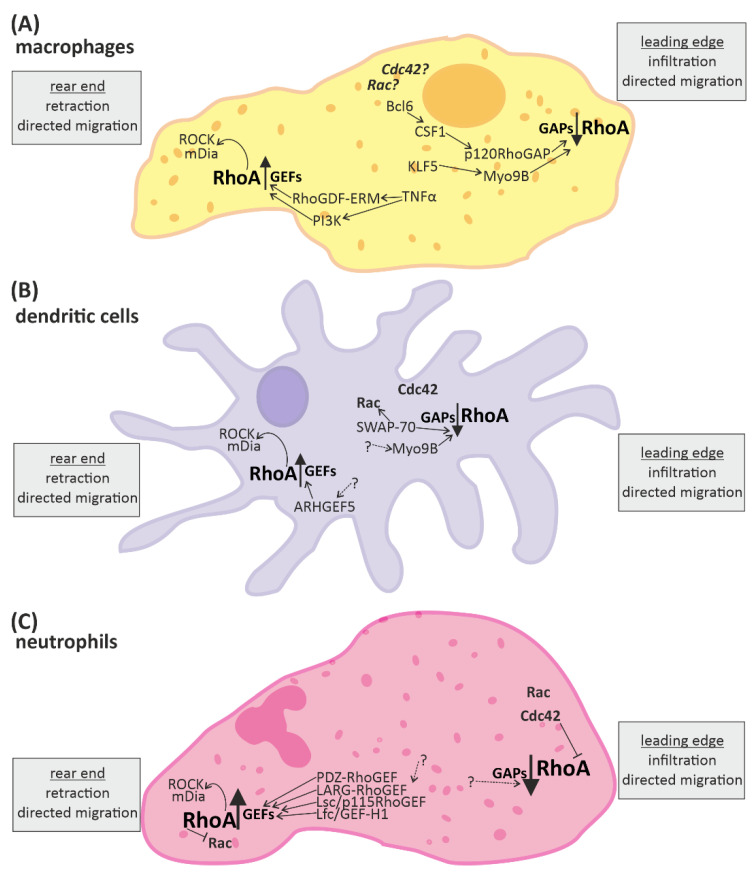
RhoA signaling plays an important role in cells of the innate immune system. Directed migration of macrophages, dendritic cells, and neutrophils is mediated by spatio-temporal regulated RhoA activation. In general, RhoA is activated in the rear end and reduced in the leading edge. The up-regulation of RhoA is mediated by “guanine nucleotide exchange factors” (GEFs) and leads to activation of the downstream effectors ROCK (rho-associated, coiled-coil-containing protein kinase) and mDia (diaphanous-related formin-1), which promote retraction of the rear end. RhoA-downregulation is mediated by “GTPase activating proteins” (GAPs) and facilitates infiltration or protrusion. Some signaling molecules up- and downstream of RhoA activation have been identified in different immune cells. However, many aspects of the signaling pathways are not fully known yet (indicated by dotted arrows and question marks). (**A**) In macrophages some regulators of RhoA have been identified. KLF5 (Krüppel-like factor-5) and CSF1 (macrophage colony stimulating receptor 1) activation via Bcl6 (B-cell lymphoma-6) lead to activation of the RhoA-GAPs Myo9B and p120RhoGAP, respectively, reducing RhoA activation at the front. Contrarily, the RhoGDF ERM and PI3K (phosphoinositid-3-kinase) promote the activation of RhoA at the end. (**B**) In dendritic cells the RhoA-GAP Myo9B is also located at the leading edge, attenuating RhoA activation, while the Rho-GEF5 is found in the rear end, promoting RhoA activation. Furthermore, SWAP-70 (switch-associated protein 70) was identified as a regulator of small GTPase activity. (**C**) In neutrophils a number of GEFs facilitating RhoA activation have been identified. In addition the small GTPases Rac and Cdc42 are located close to the leading edge.

**Figure 3 cells-10-01681-f003:**
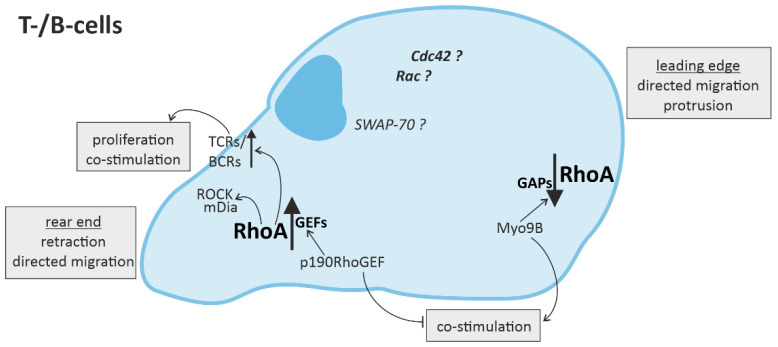
RhoA signaling in cells of the adaptive immune system. Directed migration of T- and B-cells is regulated by spatio-temporal activation of RhoA, with mainly RhoA activation in the rear end and RhoA downregulation in the leading edge. In T- and B-cells, RhoA activation in the rear end not only leads to retraction via ROCK (rho-associated, coiled-coil-containing protein kinase) and mDia (diaphanous-related formin-1), but is also associated with the promotion of T-cell receptor (TCR) or B-cell receptor (BCR) activity, respectively, inducing T-cell proliferation and co-stimulation of more immune cells.

## Data Availability

Not applicable.

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
