# Peer review of "RhoA Signaling in Immune Cell Response and Cardiac Disease"

_cells, 2021, doi:10.3390/cells10071681_

Round 1
Reviewer 1 Report
This is a brief review on the role of RhoA in cardiac pathophysiology, in particular inflammatory signaling known to contribute importantly to myocardial damage, hypertrophy and fibrosis. The article is from a team with good research contribution in this field, and in general well written. The contents of the article are well considered. This article is potentially interesting to Cells readers.
Comments:
- Regarding signaling that activates RhoA, in addition to necrosis, please consider additional description of “programmed necrosis”, either necroptosis or ferroptosis, that is increasing regarded as a key factor driving the development of heart disease.
- Beta2-AR and RhoA signaling: sympathetic nervous system is activated in the setting of heart disease and pro-inflammatory. In this regard, inflammatory cells are known to be equipped mainly with beta2-adrenergic receptors. There are studies showing b2AR-RhoA signaling (see: PMID22500016, PMID12972572). This is relevant to this review and should be included.
- RhoA and rear-front signaling (Fig 3): this is recent progress in inflammatory infiltration in which RhoA signaling plays a key role. Description of this concept, however, deserves further improvement. The current diagram failed in delivering the phenomenon of Rear-Front (ie. uropod-pseudopod) signaling leading to asymmetric force generation at the trailing edge of immune cell. Also, please cite a review article by Hind LE et al: Developmental Cell 2016;38(2):161-9
- Figures: Fig 1 can be omitted because its content is better presented in Fig 2; Fig 4A and B can be combined as one due to virtually lack of difference, except TCR or BCR.
- Perspectives: here is the place to further emphasize experimental studies testing effect of either specific RhoA gene-targeting or regulatory molecules in the setting of heart disease. This reviewer suggest that this deserves a summary table.
- Many abbreviations are used but without description.
Author Response
Reviewer 1
Comments:
- Regarding signaling that activates RhoA, in addition to necrosis, please consider additional description of “programmed necrosis”, either necroptosis or ferroptosis, that is increasing regarded as a key factor driving the development of heart disease.
Answer: We considered the reviewer’s suggestion and added a paragraph about the role of necroptosis in heart diseases and its association with RhoA signaling.
- Beta2-AR and RhoA signaling: sympathetic nervous system is activated in the setting of heart disease and pro-inflammatory. In this regard, inflammatory cells are known to be equipped mainly with beta2-adrenergic receptors. There are studies showing b2AR-RhoA signaling (see: PMID22500016, PMID12972572). This is relevant to this review and should be included.
Answer: Following the reviewer’s advice we included a paragraph describing the role of the sympathetic nervous system, b2AR-activation and RhoA signaling in the context of heart failure and additional information about b2AR-activation in immune cells.
- RhoA and rear-front signaling (Fig 3): this is recent progress in inflammatory infiltration in which RhoA signaling plays a key role. Description of this concept, however, deserves further improvement. The current diagram failed in delivering the phenomenon of Rear-Front (ie. uropod-pseudopod) signaling leading to asymmetric force generation at the trailing edge of immune cell. Also, please cite a review article by Hind LE et al: Developmental Cell 2016;38(2):161-9
Answer: Following the reviewer’s suggestion we improved the description and discussion of RhoA signaling in neutrophils mentioning rear-front signaling and asymmetric force generation. For this we included the paper recommended by the reviewer and other additional literature. Furthermore, we revised the corresponding figure.
- Figures: Fig 1 can be omitted because its content is better presented in Fig 2; Fig 4A and B can be combined as one due to virtually lack of difference, except TCR or BCR.
Answer: As the reviewer suggested, we omitted the previous figure 1, but in addition we included the scheme of different cardiac diseases converging in myocardial damage in the previous figure 2 (now Figure 1). As also suggested, we combined the two parts of the previous figure 4A+B with T- and B-cell signaling in one (now Figure 3).
- Perspectives: here is the place to further emphasize experimental studies testing effect of either specific RhoA gene-targeting or regulatory molecules in the setting of heart disease. This reviewer suggest that this deserves a summary table.
Answer: We followed the reviewer’s advice and added more information about recent/current studies regarding therapeutic approaches related to RhoA activation in the “perspectives” section.
- Many abbreviations are used but without description.
Answer: Where definitions of abbreviations were missing, we added them.

Reviewer 2 Report
This manuscript by Kilian et al, is a review article focusing on the role of the RhoA signaling in immune cell response and cardiac diseases. This is indeed a fascinating area of research and topic of interest at the interface of basic (and mostly conserved) cellular mechanisms of RhoA signaling in immune cells and the physiological consequences in heart disease. I laud author’s effort in identifying this poorly understood topic and with only a few comprehensive reviews. In this current manuscript, authors have organised the manuscript into 3 broad sections following a brief introduction. First (Section 2), authors highlight the links between cardiac ailment and RhoA signaling primarily focussing on the contrasting pro- and anti-inflammatory roles played my macrophages. Next (Section 3), authors move on to describe the regulation of RhoA signaling in different subsets of immunes cells involved in the two arms of immune system, innate and adaptive. Finally, authors summarize the open questions in the field as Perspectives (Section 4). I like the overall organisation of the review, however the same set of authors have published a very recent review on a similar topic titled:
‘RhoA: a dubious molecule in cardiac pathophysiology’ in the Journal of Biomedical Science (https://doi.org/10.1186/s12929-021-00730-w). It becomes very hard to justify the novelty and uniqueness of the current manuscript in Cells, which comes on the heels of this earlier publication. My concerns are detailed below, and I hope authors make a sincere effort to address and clarify these concerns.
Major Concerns:
- Authors need to clarify how this current review article submitted to Cells is uniquely different from their recent review in J Biomed Sci. Both reviews focus on the role of RhoA in cardiac disease. While the J Biomed Sci review adds a few more sections on RhoA superfamily, post translational modifications and regulation, there are several overlapping sections between the two reviews. for eg ‘Links between RhoA and immune system in cardiac hypertrophy’ appears in both reviews with overlapping source material. It’s important to define how this submitted manuscript complements the other recently published one, instead of duplicating the same material across two review articles.
- Apart from the concerns raised above (point 1); the sections focussing on RhoA signaling and its regulation (Section 3.1 & 3.2) have strong overlaps with another review published in Cells in 2019 from Bros et al. ‘RhoA as a Key Regulator of Innate and Adaptive Immunity’ see 10.3390/cells8070733. This 2019 Cells review (cited in the current MS as Ref. 13) has been rather frequently cited in the current manuscript, in a manner it seems to this reviewer that authors have almost based the sections and material in this manuscript from this earlier review (Ref. 13). This is specifically true for sections on cell-specific RhoA signaling across immune cells. In fact, there are several instances where authors have cited the Cells 2019 review (Ref. 13) instead of citing primarily literature to support their statements. A review should ideally highlight the original papers and not just other existing reviews. I request authors to revise the current manuscript and cite primary literature where necessary. I’ll provide some of these instances in next comments.
Other concerns:
- In Line 210, authors mention ‘PI3K and Rho-GDF ERM’. Please define what GDF is before using the acronym. Also Ref. 52 cited here is a review and it will be good to refer and cite the primary literature. In fact, Ref. 52 doesn’t even talk about Rho-GDF ERM at all. The correct reference here is K Takahashi et al, J Biol Chem, 1997 (DOI: 10.1074/jbc.272.37.23371).
- In Line 230 onwards, the discussion on RhoA activity is neutrophil needs to be elaborated to reflect the previous work done in the field. RhoA activity is localised to the uropod and Anna Huttenlocher has an excellent review (10.1016/j.devcel.2016.06.031) which I recommend authors to look at for potential sources of primary literature to bolster this section. Once again, authors cite Ref 13 (line 231) to summarise the data for neutrophil in spite of other equally or more relevant reviews in the field.
- On the same note, the schematics in Fig 3 need to be changed for Macrophage (a) and Neutrophils (c). The shapes of the cells and their polarity need to be flipped to reflect the wider more spread out part of the cell as the front/leading edge and the narrower bud like end of the cell as the uropod or rear-end. This morphology of the neutrophils is well known from several studies and authors also refer to the same in their text. Similarly, the schematics for T-cells and B-cells in Fig 4A can be flipped to indicate the correct shape and polarity of these cells.
- I like the Perspectives section put together by the authors. I wonder if it’s possible to highlight these unknown questions in a bullet point fashion. It will allows readers to carry home the distilled message of the entire review.
Author Response
Reviewer 2
Major Concerns:
- Authors need to clarify how this current review article submitted to Cells is uniquely different from their recent review in J Biomed Sci. Both reviews focus on the role of RhoA in cardiac disease. While the J Biomed Sci review adds a few more sections on RhoA superfamily, post translational modifications and regulation, there are several overlapping sections between the two reviews. for eg ‘Links between RhoA and immune system in cardiac hypertrophy’ appears in both reviews with overlapping source material. It’s important to define how this submitted manuscript complements the other recently published one, instead of duplicating the same material across two review articles.
Answer: Our previous review, which is mentioned above, indeed includes a paragraph about the “links between RhoA and the immune system in cardiac hypertrophy” in general, but this is only a compendious paragraph in addition to the comprehensive sections about RhoA activation and the many functions of RhoA in cardiac pathophysiology described there. The present review however, focuses on RhoA signaling in immune cells, its role in immune system responses and the associations with cardiac diseases and describes this in much more detail especially describing RhoA signaling in a number of immune cells (macrophages, dendritic cells, neutrophils, T- and B-cells), highlighting similarities and differences. An overlap of literature cited is a matter of course. Nevertheless, we took the reviewer’s concern into account. By improving the manuscript according to all the suggestions made by both the reviewers (especially including more details about rear-front signaling and RhoA signaling in neutrophils and more primary literature), as well as the arguments stated above we believe we addressed this concern satisfactory.
- Apart from the concerns raised above (point 1); the sections focussing on RhoA signaling and its regulation (Section 3.1 & 3.2) have strong overlaps with another review published in Cells in 2019 from Bros et al. ‘RhoA as a Key Regulator of Innate and Adaptive Immunity’ see 10.3390/cells8070733. This 2019 Cells review (cited in the current MS as Ref. 13) has been rather frequently cited in the current manuscript, in a manner it seems to this reviewer that authors have almost based the sections and material in this manuscript from this earlier review (Ref. 13). This is specifically true for sections on cell-specific RhoA signaling across immune cells. In fact, there are several instances where authors have cited the Cells 2019 review (Ref. 13) instead of citing primarily literature to support their statements. A review should ideally highlight the original papers and not just other existing reviews. I request authors to revise the current manuscript and cite primary literature where necessary. I’ll provide some of these instances in next comments.
Answer: By implementing the suggested improvements given by both reviewers, especially adding more information from different reviews and more primary literature in the sections mentioned above, we believe we addressed this concern satisfactory, too.
Other Concerns:
- In Line 210, authors mention ‘PI3K and Rho-GDF ERM’. Please define what GDF is before using the acronym. Also Ref. 52 cited here is a review and it will be good to refer and cite the primary literature. In fact, Ref. 52 doesn’t even talk about Rho-GDF ERM at all. The correct reference here is K Takahashi et al, J Biol Chem, 1997 (DOI: 10.1074/jbc.272.37.23371).
Answer: GDF was already defined before (line 184 of manuscript1). However, we added a brief description of the regulators of RhoA activation including GDFs. Furthermore, we corrected the reference as pointed out by the reviewer.
- In Line 230 onwards, the discussion on RhoA activity is neutrophil needs to be elaborated to reflect the previous work done in the field. RhoA activity is localised to the uropod and Anna Huttenlocher has an excellent review (10.1016/j.devcel.2016.06.031) which I recommend authors to look at for potential sources of primary literature to bolster this section. Once again, authors cite Ref 13 (line 231) to summarise the data for neutrophil in spite of other equally or more relevant reviews in the field.
Answer: Following the reviewer’s suggestion we revised the section on RhoA activity in neutrophils and included additional information from the recommended review as well as other literature (see also answer to comment 3 of reviewer 1).
- On the same note, the schematics in Fig 3 need to be changed for Macrophage (a) and Neutrophils (c). The shapes of the cells and their polarity need to be flipped to reflect the wider more spread out part of the cell as the front/leading edge and the narrower bud like end of the cell as the uropod or rear-end. This morphology of the neutrophils is well known from several studies and authors also refer to the same in their text. Similarly, the schematics for T-cells and B-cells in Fig 4A can be flipped to indicate the correct shape and polarity of these cells.
Answer: We revised the shape of the cells depicted in the corresponding figures, following the reviewers comment.
- I like the Perspectives section put together by the authors. I wonder if it’s possible to highlight these unknown questions in a bullet point fashion. It will allow readers to carry home the distilled message of the entire review.
Answer: We thank the reviewer for this positive feedback. We refrained from converting the statements from text to a list, however, as we believe the context to be easier to understand by the readers in the given way.

Round 2
Reviewer 2 Report
Authors have addressed most of my earlier concerns to the best of their ability. The revised version has certainly improved with the addition of the new text and references. The figures have also been corrected to reflect the representative polarity of these cells. I have a few minor suggestions on this version for authors to consider.
- The paragraph starting Line 368 onwards, does a nice summary of the underlying spatio-temporal dynamics of RhoA regulation and cross-talk in polarity and migration, which is nicely captured in the statement on line 372-373. I recommend adding one more sentence here, something on the lines to generalize the observations seen in immune cells are conserved across other cellular process. Something like this 'Recent studies have also now suggested Rho-GTPase signaling pathways crosstalk with each other and are influenced by cellular mechanics, leading to self-organisation of the several dynamic cellular processes. (refer to following reviews like PMID: 29632270, 31999511, 27533896).' This statement can be followed by the concluding sentence ' Further research....diseases'.
I recommend an acceptance once the above are addressed.
Author Response
Dear Reviewer,
Thank you very much for your favorable comments and recommending our manuscript. We have added a sentence and appropriate references (highlighted in yellow) to the desired section as per your suggestions.
Kind regards,
Ashraf Rangrez